# Physical Activity in Long COVID: A Comparative Study of Exercise Rehabilitation Benefits in Patients with Long COVID, Coronary Artery Disease and Fibromyalgia

**DOI:** 10.3390/ijerph20156513

**Published:** 2023-08-03

**Authors:** Claire Colas, Yann Le Berre, Marie Fanget, Angélique Savall, Martin Killian, Ivan Goujon, Pierre Labeix, Manon Bayle, Léonard Féasson, Frederic Roche, David Hupin

**Affiliations:** 1Department of Clinical and Exercise Physiology, University Hospital Center of Saint-Etienne, 42000 Saint-Etienne, Francedavid.hupin@chu-st-etienne.fr (D.H.); 2INSERM, U1059, DVH Team, SAINBIOSE, Jean Monnet University, 42000 Saint-Etienne, France; 3Jacques Lisfranc Medicine Faculty, Jean Monnet University, 42000 Saint-Etienne, France; 4Department of Education and Research in General Practice, Jean Monnet University, 42000 Saint-Etienne, France; 5Department of Internal Medicine, University Hospital Center of Saint-Etienne, 42000 Saint-Etienne, France; 6CIRI—Centre International de Recherche en Infectiologie, Team GIMAP, Jean Monnet University, Claude Bernard Lyon 1 University, Inserm, U1111, CNRS, UMR530, 42000 Saint-Etienne, France; 7CIC 1408 Inserm, University Hospital Centre of Saint-Etienne, 42000 Saint-Etienne, France; 8Inter-University Laboratory of Human Movement Biology, EA 7424, Jean Monnet University, 42000 Saint-Etienne, France

**Keywords:** long COVID, exercise, rehabilitation, secondary prevention, comparative study, coronary artery disease, fibromyalgia, post-exertional symptom exacerbation

## Abstract

Exercise in long COVID is poorly studied. Nevertheless, exerciserehabilitation could improve cardiorespiratory, muscular and autonomic functions. We aimed to investigate improvement in physical and autonomic performances of long COVID patients (*n* = 38) after a 4-week exercise rehabilitation program (3 sessions/week) compared to two control groups composed of coronary artery disease (*n* = 38) and fibromyalgia patients (*n* = 38), two populations for whom exercise benefits are well known. Efficacy of exercise training was assessed by a cardiopulmonary exercise test, a handgrip force test, and a supine heart rate variability recording at rest before and after the rehabilitation program. Cardiorespiratory and muscular parameters were enhanced after exercise rehabilitation in the three groups (*p* < 0.001). No significant difference was observed for the autonomic variables. Through this comparative study with control groups, we confirm and reinforce the interest of caring for long COVID patients without post-exertional symptom exacerbation by exercise rehabilitation of both strength and endurance training, by personalizing the program to the patient and symptoms.

## 1. Introduction

Severe acute respiratory syndrome coronavirus 2 (SARS-CoV-2) began in the spring of 2020 and has infected more than 600 million humans worldwide [1], with 34 million positives cases confirmed in France [2]. In the months following the first wave of COVID-19, many patients experienced persistent symptoms after being infected. These post-acute sequelae of COVID-19, also known as “long COVID” have been defined as the persistence of symptoms (new or developed following infection) beyond four weeks of an acute COVID-19 infection and could affect patients who had presented a mild or asymptomatic form [1,3,4]. The most frequently reported symptoms were asthenia, dyspnea, cognitive disorders and decrease in sports performance [4,5,6]. Long COVID can persist from a few weeks to several months or even years [7]. Indeed 36% of COVID-19 survivors presented at least one symptom between 3 and 6 months after diagnosis [8]. 

For many years, the benefits of regular physical activity have been described for cardiovascular, pulmonary and metabolic functions [4,9,10]. In primary prevention, Steenkamp et al. (2022) showed that moderate to vigorous physical activity before being infected with COVID-19 significantly reduced the risk of hospitalization, the need for intensive care, mechanical ventilation necessity, and death [11]. However, exercise in secondary prevention has been less investigated in these situations of symptoms persistence. A scoping review of long COVID exercise rehabilitation by Prado et al. (2022) showed that there were only a few experimental studies and no randomized controlled trials on the topic, thus weakening the conclusions on the effectiveness of exercise as a mean to rehabilitate post-COVID-19 [12]. However, a number of studies provided recommendations for exercise rehabilitation which were mostly extrapolated from the management of other chronic conditions such as cardiac and/or pulmonary disease [12,13,14]. The exercise rehabilitation program must include both aerobic and strength workouts, at least for 4 weeks with 2–5 sessions per week of at least 30 min each. 

Exercise in long COVID could be relevant to limit the potential consequences of the pathology both in terms of symptoms (e.g., fatigue [15] and breathlessness [16]) and exercise intolerance [17]. More generally, the autonomic alteration described in these patients could be counterbalanced by regular physical activity [18]. However, special attention should be given to patients showing autonomic dysfunction [19]. Sometimes observed in long COVID and described in others somatic disorders, post-exertional symptom exacerbation (PESE) is characterized by worsening of symptoms following physical or mental exertion, generally 12–48 h after the activity and lasting for several days or (rarely) weeks [20]. For these patients with autonomic dysfunction, a symptom-titrated exercise protocol is necessary or even a contraindication to exercise for patients whose assessment tests have detected PESE [19,20,21].

The lack of hindsight on the effectiveness of an exercise rehabilitation program in patients with long COVID makes it difficult to interpret the results, especially as most of the available studies were not controlled with other pathological subjects or even healthy subjects. To assess the efficacy of exercise rehabilitation in long COVID, it seems pertinent to analyze its benefits compared to other patients for whom exercise rehabilitation is an integral part of care. Exercise rehabilitation programs for long COVID are inspired by cardiopulmonary programs. In this sense, the inclusion of a cardiac pathology group as a control for the effectiveness of exercise rehabilitation makes perfect sense. Exercise benefits in cardiovascular pathologies are widely documented in the literature [22,23]. However, they are rather distant from long COVID forms and a comparison with a more similar pathology group would be warranted. Long COVID is a clinically unspecific syndrome and has been related to functional somatic disorders [24,25]. Among these functional somatic disorders, fibromyalgia is characterized by multiple subjective symptoms, showing many similarities with long COVID [26]. For these patients, symptom-titrated exercise is the first therapy-based recommendation [27,28]. In both groups of patients, exercise in secondary prevention reduced pain and fatigue, and improved functional capacities and quality of life [29,30,31].

The evaluation of exercise rehabilitation efficacy is based on a cardiopulmonary exercise test (CPET) before and after the exercise rehabilitation program [12]. CPET is both well-tolerated and safe in post-COVID-19 [32]. Peak oxygen uptake (VO_2_ peak) is the main prognostic marker of morbidity and mortality [22], and there is a correlation between this value and quality of life in healthy subjects and patients [33,34]. Considering that patients are not always able to achieve a maximum effort, the ventilatory threshold should be considered as a better fitness index [35], and it allows for the individualization of the exercise training program to each patient’s own training capacity and for the objective effectiveness [36]. Also, an evaluation of autonomic function through heart rate variability (HRV) parameters for the long COVID population would be relevant. HRV is widely used to analyze the evolution of symptoms of stress, sleep apnea, fatigue, and their impact on several pathologies [37,38]. HRV data are obtained by analyzing the electrocardiogram signal in both time and frequency domains, in order to observe the sympathovagal balance. In the resting condition, the square root of the mean squared differences of successive RR intervals (RMSSD) and high frequency (HF) index reflect the parasympathetic system, while the low frequency (LF) primarily reflects sympathetic activity but is influenced by baroreceptors activity and parasympathetic tone. LF/HF ratio could estimate the ratio between sympathetic/parasympathetic system, and thus reflects autonomic balance [39,40].

To assess the efficacy of exercise rehabilitation in long COVID, we investigated the improvement in physical and autonomic performances of patients with persistent symptoms following a one-month rehabilitation program. These results were compared to performances of two other groups with chronic diseases, more precisely a first control group composed of coronary patients and a second control group composed of fibromyalgia patients. We aimed to compare the responses of these three patient profiles to an exercise program. Our hypothesis was that, as for coronary and fibromyalgia patients, exercise rehabilitation will improve physical performance in patients with long COVID.

## 2. Materials and Methods

### 2.1. Study Design 

This was a prospective comparative study which included 38 long COVID patients from the COVIMOUV study (ClinicalTrials.gov Identifier: NCT05236478) compared to two control groups composed of patients with coronary artery disease from the CITIUS study (ClinicalTrials.gov Identifier: NCT04102410, *n* = last 38 patients included and randomized in exercise rehabilitation protocol) and with fibromyalgia from the FIMOUV study (ClinicalTrials.gov Identifier: NCT03736733, *n* = last 38 patients included and randomized in exercise rehabilitation protocol). These patients were enrolled in a combined prospective study and had a retrospectively gathered outcome analysis. All patients participated in a 4-week exercise rehabilitation program. They provided written informed consent before beginning the experimentation. The study was in accordance with the Declaration of Helsinki and the protocol was approved by the Ethics Committee of the university hospital of Saint-Etienne, France: IRBN142021/CHUSTE for COVIMOUV, by the institutional review boards for CITIUS (CPP Est IV, France, IDRCB 2018-A01613-52) and FIMOUV (CPP Sud II, France, IDRCB 2019-A02221-56).

Inclusion criteria were the following: patients over 18 years old; previous infection with SARS-CoV-2 confirmed by reverse transcription polymerase chain reaction, hospitalization or not and showing persistent asthenia > 3 months for long COVID patients; acute coronary syndrome treated within the last 6 months for coronary patients; and diagnosis made according to 2016 American College of Rheumatology criteria for fibromyalgia patients. Patients with comorbidities preventing the practice of physical activity including those with reported symptoms of PESE at the initial assessment were excluded. 

### 2.2. Exercise Rehabilitation Program 

Long COVID patients benefited from a multidisciplinary physical rehabilitation protocol over four weeks at the rate of three 2 h sessions per week, with two sessions of exercise (consisting of 90 min of aerobic exercise and 30 min of resistance exercise) and one therapeutic education session on symptom management (fatigue, sleep, nutrition and exercise). The two control groups followed an exercise rehabilitation adapted to the recommendations specific to each population. The coronary artery disease patients participated in five sessions of 2 h of cardiac rehabilitation (including one of therapeutic education per week) for four weeks. The rehabilitation protocol for fibromyalgia patients included two exercise sessions of 90 min and one therapeutic education workshop per week over four weeks. All exercise rehabilitation programs were attended in person and supervised by qualified professionals, i.e., an adapted physical activity instructor and/or physiotherapist.

### 2.3. Assessment Measures

Outcomes assessment was performed at baseline (PRE) and at the end of the 4-week intervention (POST). Outcomes were used in order to assess the efficacy of exercise training and concerned the following: Aerobic performances: improvement in VO_2_ peak, VO_2_ at the first ventilatory threshold (VT1) and maximal aerobic power (MAP) were assessed by a CPET (Vyntus CPX, CareFusion, San Diego, CA, USA);Anaerobic performances: improvement in upper limb muscular strength was assessed by a handgrip test (in kg) with a Jamar hydraulic hand dynamometer (JLW instruments, Chicago, IL, USA);Cardiac autonomic nervous system (ANS) performances: improvement in resting heart rate (HR), baroreflex sensitivity (BRS) and HRV throughout RMSSD, LF and HF indexes and LF/HF ratio were assessed from a 15′ resting heart recording in the supine position (Finapres Medical Systems BV, Enschede, The Netherlands). BRS assesses the ability to regulate fluctuations in blood pressure and HR through the activation of the sympathetic and orthosympathetic branches of the ANS. Impaired BRS is associated with numerous pathologies, notably cardiovascular, and its reduction is a predictive factor for cardiac mortality [41]. Physiological values of HRV for short-term recording in healthy population have been described by Nunan et al. [42]. Altered ANS in patients is generally characterized by lower parasympathetic activity (mainly measured by RMSSD and HF) and higher sympathetic activity (mainly measured by LF) compared to healthy controls; the LF/HF ratio is therefore higher overall and indicating sympathetic predominance [43,44]. These values were expressed as the natural logarithm to correct the skewed distribution [40].

### 2.4. Statistical Analysis

Statistical analyses were performed using R (R Development Core Team 2020, Vienna, Austria) and Jamovi statistical software (version 2.2.5). Data were expressed as mean (±standard deviation) and frequencies (%). One-way ANOVA tests were used to compare groups at baseline. To compare the evolution of the three groups, a mixed regression model adjusted for age, sex and smoking (significantly different between groups at baseline) were performed to determine statistically significant changes. Effect sizes were determined through the estimated coefficient of the model, corresponding to the average POST-PRE value for the three groups, in the unit of measurement as the response variable, corrected for the confounding variables added in the model. A Bonferoni post hoc test was performed when a time*group interaction was found (VO_2_ peak). The level of significance was set at *p* < 0.05.

## 3. Results

In total, 38 patients with long COVID (55% women, mean age 46.9 ± 12.7 y) were recruited and compared to 38 coronary artery patients (24% women, mean age 61.4 ± 9.45 y) and 38 fibromyalgia patients (92% women, mean age 47.4 ± 9.93 y) (Figure 1). 

Significant differences between groups were observed at baseline for age, sex and tobacco consumption (*p* < 0.05) (Table 1). Long COVID patients were 50% overweight with a mean body mass index (BMI) of 25.4 ± 5.44 kg·m^−2^. Patients who practiced physical activity before COVID-19 infection were 66%. Most frequent persistent symptoms concerned asthenia (100%), exertional dyspnea (84%, *n* = 32), sleep disorder (50%, *n* = 19), cognitive impairment (47%, *n* = 18), diffuse pain (39%, *n* = 15), anxiety syndrome (24%, *n* = 9), and chest pain (18%, *n* = 7). Tachycardia, cough and digestive disorders were each reported by <5% (*n* = 2).

Significant baseline differences were found for all parameters except BRS, showing higher aerobic capacities in the COVID group and lower anaerobic capacities in the fibromyalgia group compared to the other two groups. The difference in resting HR could be partly explained by the bradycardia-inducing treatment in all coronary patients. Results are presented in Table 2. 

Except for VO_2_ peak, cardiorespiratory and muscular parameters were similarly improved after the exercise rehabilitation program in the three groups. Indeed, VO_2_ at VT1, MAP and handgrip strength significantly increased at the end of the exercise rehabilitation program for the three groups (*p* < 0.001), without significant time-by-group interaction (*p* = 0.517, *p* = 0.756, *p* = 0.059, respectively). Regarding the VO_2_ peak, this parameter was improved (*p* < 0.001) and we found an interaction effect (*p* = 0.019) for which the post hoc test identified a significant difference between the COVID and the coronary groups after their rehabilitation (*p* = 0.046). Thus, the gain in this parameter was significantly different between the two groups (+18 ± 19% vs. +10 ± 16%, respectively).

No significant difference was observed at the end of the experimentation for the autonomic variables. Exercise rehabilitation did not improve patients’ autonomic function. 

## 4. Discussion

This comparative study aimed to assess the effectiveness of exercise rehabilitation in patients suffering from long COVID compared to coronary and fibromyalgia patients. Our results showed a similar improvement in main physical performance parameters, VO_2_ at VT1, MAP and handgrip force, demonstrating that exercise rehabilitation for long COVID patients without PESE was as effective as exercise rehabilitation for other patients. 

CPET is a tool for assessing physical condition and can be used to identify the limiting factors in the reduction of maximal effort after each phase of disease convalescence [17]. CPET has been widely used to assess cardiovascular changes during exercise in long COVID patients [45,46,47]. Studies unanimously concluded that exercise capacity is impaired [17,48]; and Durstenfeld et al. (2022) showed that exercise capacity was reduced by 4.9 mL·kg^−1^·min^−1^ among long COVID patients compared to individuals without symptoms after SARS-CoV-2 infection [49]. Our results supported this finding, with VO_2_ peak values assessed at 23.0 ± 5.92 mL·min^−1^·kg^−1^ and VO_2_ at threshold at 15.6 ± 5.03 mL·min^−1^·kg^−1^. These measures seemed comparable to other values reported in the literature, such as in the study by Barbaglata et al. (25.8 ± 8.1 mL·min^−1^·kg^−1^) [50] or Contreras et al. (21.9 ± 6.4 mL·min^−1^·kg^−1^) [51], considering that no differences were found according to the level of severity of COVID [52,53]. Compared to the other two populations presented here, it seems that the cardiorespiratory profile of long COVID patients is less altered. This could be explained by the relatively early care offered [54,55], which avoids the chronic state of physical deconditioning that is well described in fibromyalgia [56,57,58]. Moreover, the age, gender, and smoking profile was different because of coronary risk factors that are more prevalent in older men and smokers, whereas the long COVID and fibromyalgia population appears to be younger and female. All populations were overweight, i.e., BMI > 25 kg·m^−2^. 

Despite the potential impairment of exercise capacity in long COVID patients, CPET assessments showed that a mixed exercise rehabilitation on the model of cardiac rehabilitation was effective in improving aerobic (+18% in VO_2_ peak) and anaerobic (+17% on handgrip strength) performances after a 4-week exercise rehabilitation program. This is in accordance with the latest and rare studies on benefits of exercise in long COVID [59,60]. In particular, the study by Barbara et al. on a larger long COVID population (*n* = 50) without a control group showed an improvement in VO_2_ peak values of around 15% following an 8-week exercise rehabilitation [61]. To our knowledge, handgrip strength was not evaluated in long COVID. However, it was reduced in COVID survivors without persistent symptoms compared to obstructive sleep apnea syndrome patients, and was independently correlated to prior hospitalization due to COVID-19 [62]. The improvement in strength after exercise rehabilitation is known as a predictor of quality of life in older adults [63] as in COVID patients without persistent symptoms, justifying an exercise rehabilitation program with a focus on increasing muscle strength [64]. In our results, it seemed that the mixed exercise program improved cardiovascular and muscular parameters without detecting differences with the comparison groups; VO_2_ peak was the only parameter for which long COVID patients seemed to have a better improvement than coronary patients.

Autonomic dysfunction is a major hypothesis of the symptom’s persistency in long COVID [7,65,66]. A chronotropic incompetence or inadequate HR recovery were suggested to explain autonomic dysfunction throughout CPET values [17,67]. However, this did not allow a direct measurement of autonomic activity. HRV analysis was a complementary tool for exploring the ANS in order to objectively assess its alteration. Some authors have already shown dysautonomia in long COVID patients [68]. Carvalho Marques et al. (2022) compared the HRV of long COVID patients to those of healthy controls. Their results revealed a reduction in global HRV, increased sympathetic modulation influence and a decrease in parasympathetic modulation in long COVID [69]. A recent systematic review confirmed a decreased parasympathetic activity, although the authors suggested standard deviation of all NN intervals as a reference parameter [70]. This seems comparable to our HRV data, suggesting long COVID patients tend to have a sympathetic profile with a very low parasympathetic component. Their very low RMSSD, high resting HR and LF/HF ratio values indicated an imbalance of the ANS [37] which was not found in healthy controls [69]. The data from our study showed that this impairment was not different from that observed in other pathologies such as coronary disease or fibromyalgia in which HRV is commonly investigated [71,72,73]. For these control patients, the autonomic activity was explored and it appeared that the parasympathetic was weak [44,74] and the sympathetic demonstrated hyperactivity and hyporeactivity [73]. This is in line with our results which demonstrated low parasympathetic activity and a higher sympathetic activity. However, some studies found an autonomic unbalance but characterized by an increased parasympathetic tone in long COVID patients [75,76,77]. This difference could be explained by the numerous selection criteria for HRV analysis, and the variability of age and symptoms. These parameters strongly influenced the ANS [78,79]. The analysis method and recording time could also influence the data [80].

Exercise plays an important role in enhancing sympathovagal balance and could normalize levels of sympathetic index [81,82,83]. In our study, autonomic parameters were not improved after a 4-week exercise rehabilitation in the three groups. This is consistent with the literature on cardiovascular diseases and fibromyalgia, which did not find any improvement in HRV indexes after 8 training weeks [84,85,86]. A higher volume of training >4 months seems beneficial in cardiac [87] as well as in fibromyalgia patients [88,89]. Likewise, the study of Del Valle et al. (2022) had shown an improvement of resting HR after 8 weeks of supervised pulmonary rehabilitation on severe COVID-19 survivor patients [90], while we did not find significant improvement for this parameter in our study. The volume and duration of physical activity proposed in this study was certainly too low to expect autonomous changes.

With the multiplication of cases of long COVID, the management of multiple symptoms presented by patients has become a public health priority. Exercise rehabilitation could be a promising therapeutic approach and recent studies aimed to determine the most suitable exercise rehabilitation protocol for long COVID patients. The practice of strength and endurance exercises is found in some of them [59,60,91]. More particularly, a recent review has introduced the “new combined post-COVID-19 exercise protocol” composed of aerobic exercise and resistance training (2–5 sessions per week for 12 weeks) [13]. In our study, a shorter exercise rehabilitation program has demonstrated benefits of endurance exercises, seen above with the improvement in VO_2_ data. In addition, we demonstrated the improvement of functional capacities through an increased handgrip strength. Our mixed program was relevant since we found similar effects to those obtained in coronary and fibromyalgia patients, two populations where the benefits of exercise are already widely demonstrated. Other complementary approaches, such as education [92], breathing exercises [93,94] or virtual reality [16,95], are investigated and could contribute to better improvement. 

In primary care, exercise prescription by the general practitioner in a safe environment with a qualified facilitator and in a certified structure could be a way to manage symptoms of long COVID patients. As highlighted by Greenhalgh et al. (2022), “many patients can be supported effectively in primary care” [96]. This has also been discussed in the Swedish context [97]. This prescription should be preceded by an initial consultation to detect “red flags”, e.g., comorbidities and risk of PESE, an autonomic dysfunction which may contraindicate the practice of physical activity, sometimes described as “harmful” [7,20]. In long COVID, PESE was found in 8–75% of patients according to the literature [98,99,100], which explains why some authors preferred to proscribe physical activity in long COVID [7,101], supported by the results of a recent cross-sectional study in which 75% of participants declared worsening symptoms with physical activity [102]. However, our results showed that an adapted physical activity performed in a safe, appropriate and individualized manner seems to be relevant for a subgroup of patients without PESE at baseline. For those who suffered from it, a structured pacing protocol was efficient to reduce PESE episodes and improved overall health [103]. Thus, the exercise rehabilitation program must be progressive and adapted to each patient, respecting their difficulties and limitations in order to avoid any risk of PESE [1,99,104]; this state of PESE was described as “transient” by Coscia et al. (2023) and the exploration of the signs of PESE by clinical assessment, questionnaire or CPET should be regularly performed in order to offer a suitable care [100]. To this end, the most recommended questionnaire is the DePaul Symptom Questionnaire–Post-Exertional Malaise (DSQ-PEM) [105]. For the most complex cases, patients can benefit from specialist referral if necessary.

Our study showed some limitations, the main one being the absence of a control group for long COVID patients. This did not allow us to analyze both the evolution of treated patients with the natural evolution of symptoms, which remains unknown to this day, and the effect of medical follow-up “alone”, in a dedicated structure and group. Significant differences were observed for several parameters at baseline, which suggests a noncomparability of the groups. However, the methodological choice of the last patients included in each group should not represent a selection bias a priori. Our analysis showed an improvement in objective physical parameters, but these results could be influenced by the quantity of exercise performed outside in daily life that was not assessed. The lack of any assessment of quality of life or impression of change in our study did not allow us to make a link with the clinical evolution of patients. Such an assessment would have been relevant to reinforce our results and confirm their clinical relevance. Finally, the small sample of patients in the three groups does not allow to draw definitive conclusions from this study.

## 5. Conclusions

Through this comparative study with control groups, we confirmed and reinforced the interest of caring for long COVID patients by rehabilitation in both strength and endurance exercise, and by adapting and personalizing the exercise rehabilitation program according to the patient and symptoms.

## Figures and Tables

**Figure 1 ijerph-20-06513-f001:**
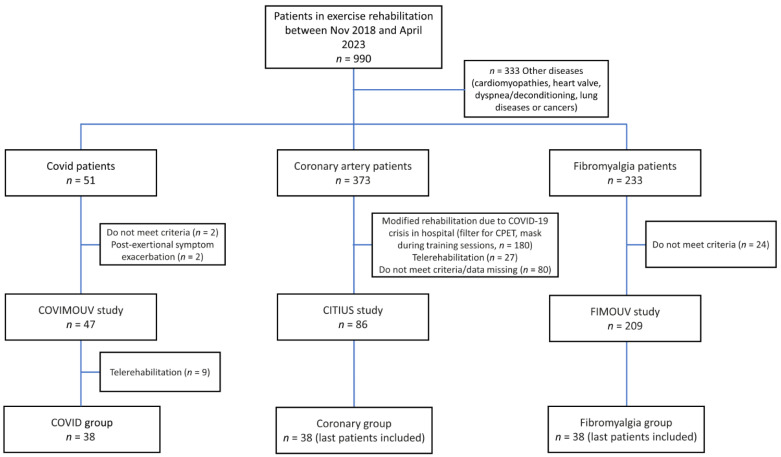
Flowchart.

**Table 1 ijerph-20-06513-t001:** Descriptive statistics of patients’ characteristics.

Variable	COVID Group (*n* = 38)	Coronary Group (*n* = 38)	Fibromyalgia Group (*n* = 38)	*p*-Value
Age (y)	46.9 ± 12.7	61.4 ± 9.45	47.4 ± 9.93	<0.001 *
Females	21 (55)	9 (24)	35 (92)	<0.001 *
BMI (kg.m^−2^)	25.4 ± 5.44	26.3 ± 4.45	28.1 ± 6.57	0.175
Physical activity	25 (66)	20 (53)	17 (45)	0.131
Tobacco	14 (37)	29 (76)	11 (29)	<0.001 *

Values are expressed as mean ± SD or n (%). BMI, body mass index. Significant differences between groups, * *p* < 0.05.

**Table 2 ijerph-20-06513-t002:** Effects of 4-week exercise training on physiological parameters in long COVID, coronary and fibromyalgia patients.

Variable	COVID Group		Coronary Group		Fibromyalgia Group		Time Effect *p*-Value	Effect Size	Group Effect *p*-Value	Time-by-Group Interaction *p*-Value
	PRE	POST	PRE	POST	PRE	POST				
**Cardiorespiratory**										
**VO_2_ at VT1** **(mL·min^−1^·kg^−1^)**	15.6 ± 5.03	17.3 ± 4.74	12.4 ± 3.49	13.3 ± 4.10	13.5 ± 3.49	15.2 ± 3.96	<0.001	1.5013	0.020	0.517
**VO_2_ peak** **(mL·min^−1^·kg^−1^)**	23.0 ± 5.92	27.0 ± 7.22 *	19.0 ± 4.52	20.4 ± 5.25 *	19.7 ± 4.83	22.8 ± 6.18	<0.001	2.928	0.039	0.019
**MAP (W)**	133 ± 50.9	154 ± 55.7	105 ± 29.9	124 ± 33.8	95 ± 36.4	112 ± 40.4	<0.001	18.711	0.031	0.756
**Muscular**										
**Handgrip force (kg)**	31.3 ± 10.1	35.7 ± 10.0	34.4 ± 9.08	36.6 ± 10.5	21.9 ± 10.6	28.7 ± 11.3 *	<0.001	4.636	0.121	0.059
**Autonomic**										
**Resting HR (bpm)**	88.8 ± 13.7	85.4 ± 11.9	69.8 ± 9.52	68.0 ± 12.8	78.6 ± 14.6	77.3 ± 10.9	0.059	−2.0925	<0.001	0.634
**BRS (ms·mmHg^−1^)**	6.06 ± 3.02	7.29 ± 3.02	7.03 ± 5.54	8.95 ± 12.3	6.78 ± 5.48	7.31 ± 4.48	0.580	1.1809	0.276	0.821
**RMSSD (ms)**	27.7 ± 12.5	31.0 ± 13.3	36.4 ± 27.9	41.8 ± 41.0	20.9 ± 16.8	23.5 ± 19.2	0.082	3.231	0.045	0.994
**LF (ms^2^) ^a^**	5.30 ± 1.06	5.30 ± 1.34	4.70 ± 1.54	5.07 ± 1.62	5.45 ± 2.05	5.44 ± 1.05	0.430	0.1367	0.231	0.510
**HF (ms^2^) ^a^**	4.43 ± 1.17	4.39 ± 1.32	4.20 ± 1.49	4.39 ± 1.55	4.81 ± 1.67	4.84 ± 1.03	0.630	0.0781	0.419	0.863
**LF/HF ^a^**	0.868 ± 0.965	1.08 ± 1.31	0.506 ± 0.749	0.679 ± 0.660	0.633 ± 0.964	0.598 ± 0.912	0.246	0.1198	0.224	0.482

PRE, initial assessment; POST, final assessment; VO_2_ at VT1, oxygen uptake at the first ventilatory threshold; VO_2_ peak, maximal oxygen uptake reached at the end of cardiopulmonary exercise test; MAP, maximum aerobic power; HR, heart rate; BRS, baroreflex sensitivity; RMSSD, square root of the mean squared differences of successive R-R intervals; LF, low frequency; HF, high frequency. Cardiorespiratory variables were obtained by a CPET; muscular variable was obtained by a force prehension on a hand dynamometer; autonomic variables were obtained by a resting heart recording. ^a^ Values were log transformed in order to correct the skewed distribution. * Post hoc test, COVID significantly different from Coronary in POST, *p* < 0.005.

## Data Availability

The raw data supporting the conclusions of this article will be made available by the authors, without undue reservation.

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
