# Peer review of "Physical Activity in Long COVID: A Comparative Study of Exercise Rehabilitation Benefits in Patients with Long COVID, Coronary Artery Disease and Fibromyalgia"

_ijerph, 2023, doi:10.3390/ijerph20156513_

Round 1

Reviewer 1 Report

 First of all, congratulations to the authors for the study. The topic is very interesting and relevant, since, as they warn in the introduction, there are not enough quality studies on this population.

HHowever, the manuscript presents some inaccuracies and drafting issues that need to be improved in order to be published and to live up to the experimental design.

Some of the observations are shown below:

Abstract
The structure of the abstract should be improved.

Introduction:

Line 48-49: Insert cite for these sentence.

Line 51-65: Include in the same paragraph the text between lines 51 and 65.

In a following paragraph, I would suggest including the rationale for the variables to be measured. -In these sense, I suggest to restructured the paragraph into three main ideas: (I) Potential consequences of long covid (autonomic dysfunction and PESE); (II) the rationale for including other groups with other pathologies as controls. In this regard, a more detailed explanation is needed to justify this choice; (III) The relevance and suitability of the CPET and HRV assessments for this population.

Line 89: If the sample did not include individuals with PESE I would discard mentioning that in the hypothesis or writing it in a declarative way.

Methods:

Line 111: Was the program attended in person? This should be highlighted as it adds value to the work.

Line 123: Why did the authors delay the evaluations for a month after the end of the protocol, and could this have affected the magnitude of the results?

Line 137: The ANS assessment protocol needs to be defined or cited and an explanation of which parameters each variable evaluates is needed.

Line 139: I suggest to include the rationale to include the statistical analysis in these paragraph.

Table 2: I would suggest (for clarity) that table 2 include a column with the p-values of the main effects and the interaction of each variable and its effect size, highlighting with * only the significant p-values in the post-hoc.

Additionally, procedure to obtain sample characteristics may be explained in this section.

Discussion

The rationalemay to be improved. Line 248 to 277 must to be reallocated into the introudction.

Line 281: explain the sample of these studies.

Line 287: Could volume alone explain the non-improvements? Maybe the protocol was too short and longer protocols need to be studied?

In summary, more detailed discussion of the evaluated variables and comparison with existing literature in the field is needed, both for the results obtained and for the lack of them.

 Moderate editing of English language required. 

Reviewer 2 Report

Thank you for the opportunity to review this manuscript. I congratulate the authors on its completion. Overall, it is well written and on a relevant and actual topic. The identification of the issues is well documented. It presents complex and innovative methodologies for the collection of the data. The reflection is guided by an adequate compilation of facts and literature.

However, I have some comments and suggestions presented below

2. Materials & Methods

Line 110. “Patients with signs of PESE at the initial assessment were excluded.” Which were the signs of PESE considered to be excluded? Some objective evaluation was made, or the testimony of the patient was the principal variable take into account? In this last case, it might be better to state that patients which reported symptoms of PESE were excluded. I think this is a relevant aspect in the selection of the sample and an innovative approach of your research, thus, I think it is adequate no clarify a little more.  

2.2. Exercise rehabilitation program

Please, can you clarify if the rehabilitation protocols were implemented in person and if some qualified professional guided the training? 

2.3. Assessment Measures

Line 132. Cardiac autonomic nervous system (ANS) performances. Is it possible for you to explain a little more about the clinical point of these variables? For example, the inclusion of the physiological values, if available, or their tendency in pathological circumstances, might be of great help to interpret the descriptive values presented in the results and as a guidance for the understanding of the discussion in this issue.  

3. Results

Line 149. Figure 1. We can observe that in the penultimate step of the figure 1, you had more that 38 patients in each group. Have you selected the last 38 patients in each group through randomization? I suppose, it was like this but you have to specify that issue previously in the 2.1. Study design subheading. 

Please, clarity a little the presentation of the results. The text is quite clear, but we don’t know if the results described in the text are or are not included in the table 2. 

For example, I think that these results are not included: “Significant baseline differences were found for all parameters except BRS, showing higher aerobic capacities in the COVID group and lower anaerobic capacities in the fibromyalgia group compared to the other two groups”, but this sentence is placed just after the mention to table 2. In the rest of the text, we can see more easily to what data in the table it refers, except for the fact that we don’t know to what comparation refers the p value in the last column of the table (I suppose that it is pre-post comparison taking the three groups together, that is, the p values of time effect, but I’m not sure.) If this is like this, it is better to specify that the time effect* in the legend of the table, it is the post-hoc time effect, for each group. 

4. Discussion

Line 189. “Our results showed a similar improvement in main cardiorespiratory parameters, i.e., VO2 and MAP”. Please, change “Our results showed a similar improvement in main physical performance parameters, VO2 at VT1, Handgrip force and MAP”

Line 221: “The rehabilitation program improved patients’ physical condition, who seemed satisfied with the care offered: long 222 COVID self-perceived satisfaction in regard to the program was reported at 7.7 ± 1.26 223 points on a 0-10 numerical rating scale.” Please, remove this sentence. I have not seen this measure in the material and methods section, and it is stated in the limitations section that the lack of this kind of evaluation is a limitation of the study. 

Line 227: “A precedent study of our research team had demonstrated a reduction in fatigue in long COVID patients who followed a 1-month hybrid rehabilitation program [41]. A meta-analysis highlighted the role of rehabilitation in reducing fatigue in patients with post-COVID-19 syndrome, with only 17% of 230 subjects reported the persistency of symptoms (n=65) [42].” You can maybe also remove these sentences from the text. You have not measured the fatigue either.

Line 255: “This assessment is generally investigated in coronary and fibromyalgia patients, why it was relevant to evaluate it in long COVID patients.” Please, syntax review. 

Line 305. “DSQ-PEM”. Please, explain the acronym. 

Limitations. Can you add a limitation about the origin of the sample. That is, all the patients have been previously in exercise rehabilitation. Do you think that this fact might have affected the outcomes in some way? For example, they were more prone to comply with the physical activity program. Or, the different quantity of physical activity performed previously could have influenced the great differences observed in the dependent variables at baseline. 

The English is ok, only minor corrections are needed. 

Round 2

Reviewer 2 Report

I thank the authors the effort made to ameliorate the manuscript. I think that the issues proposed have been correctly modified. Only, a comment: in the future, I think it is better that the professional that supervises the exercise program will be physiotherapist. This will allow to integrate the exercise program with passive techniques of physiotherapy, when necessary, for example bronchial clearance techniques or passive mobilizations of the thorax.  Congratulations for your work.